# Neural Activation in Bilinguals and Monolinguals Using a Word Identification Task

**Alejandro E. Brice** [1,*] **, Christina Salnaitis** [2] **and Megan K. MacPherson** [3]

[1] Language, Literacy, Ed.D., Exceptional Education, and Physical Education (LLEEP), College of Education, University of South Florida, Tampa, FL 33620, USA

[2] Psychology Department of Psychological Science, Kennesaw State University, Kennesaw, GA 30144, USA; csalnait@kennesaw.edu

[3] Communication Sciences and Disorders Health Professions, Central Michigan University, Mt. Pleasant, MI 48859, USA; macph1mk@cmich.edu

[*] Correspondence: aebrice@usf.edu

**Abstract:** The study investigated word recognition during neural activation in monolinguals and bilinguals. We specifically examined word retrieval and blood-oxygenation changes in the prefrontal cortex during a code-mixed word recognition task. Participants completed a gating task incorporating monolingual sentences and Spanish-English code-mixed sentences while using functional near-infrared spectroscopy (fNIRS) to measure blood-oxygenation changes. Word recognition contained four phonotactic conditions: (1) voiceless initial consonants, (2) voiced initial consonants, (3) CV-tense words, and (4) CV-lax words. Bilingual speakers had word-recognition capabilities similar to monolingual speakers even when identifying English words. Word recognition outcomes suggested that prefrontal cortex functioning is similar for early age of acquisition (AOA) bilinguals and monolinguals when identifying words in both code-mixed and monolingual sentences. Monolingual speakers experienced difficulty with English-voiced consonant sounds; while bilingual speakers experienced difficulties with English-lax vowels. Results suggest that localization of speech perception may be similar for both monolingual and bilingual populations, yet levels of activation differed. Our findings suggest that this parity is due to early age of acquisition (AoA) bilinguals finding a balance of language capabilities (i.e., native-like proficiency) and that in some instances the bilingual speakers processed language in the same areas dedicated to first language processing.

**Keywords:** speech perception; bilingual; English Spanish; code switching; code mixing; gating; fNIRS; age of acquisition (AOA); early bilinguals; phonotactics; blood oxygenation





## 1. Introduction

The purpose of this study was to investigate neurotypical responses in both monolingual and bilingual populations via a speech perception word recognition task and neural activation monitoring. Few studies have investigated the perception of words by bilingual subjects in both of their languages and when code-mixing (Fricke et al. 2019; Li 1996; Navarro-Torres et al. 2021; Peltola et al. 2012). Hernandez (2009) found that alternating between languages leads to brain activation in areas associated with executive control and articulatory/motor planning. Early age of acquisition (AoA) bilinguals may represent their two languages via neural systems not typically associated with language (Hernandez 2009). Further research investigating neural activation in populations during bilingual tasks will yield a broader understanding of where bilingual speakers may process information contrasted with monolingual speakers.

The research on bilingual speech perception has been incomplete to date as substantially more research has been conducted in speech production (López 2012; Grosjean 1996; Li 1996; Sebastián-Gallés et al. 2005; Edwards and Zampini 2008). Another issue has been the complexity of bilingualism and second language learning. Multiple factors affect

second language learning such as age of acquisition, the amount of time using a second language, and the phonetic features of both languages (Brice et al. 2021; Granena and Long 2012; Sebastián-Gallés et al. 2005). In addition, exploration of the differences between speech sound inventories of monolinguals and bilinguals has been limited (Hernandez 2009). Hence, understanding the different sound systems (i.e., phonetics of each language) may yield a better understanding of second language learning. It is documented that most bilingual speakers voluntarily engage in code-mixing in conversation (Gollan and Ferreira 2009) and bilinguals use speech sound inventories from both their first (L1) and second language (L2) when code-mixing (Genesee 2015; Grosjean 1996, 2001, p. 235; Grosjean and Miller 1994). Code-mixed language offers a unique opportunity to study first- and second-language speech perception while both languages are being assessed (López 2012). In addition, it is possible to investigate neural activation monitoring during such a task with the use of functional near-infrared spectroscopy (fNIRS). fNIRS is a non-evasive imaging technology that has been previously used to investigate bilingualism and brain localization function (Jasinska and Petitto 2013; Kovelman et al. 2008, 2014; Zinszer et al. 2015).

*1.1. Bilingual Speech Perception*

The development of speech production and perception occurs within the constraints of the phonetic inventory and vernacular of the language being learned (Anthony et al. 2009; Birdsong 2018; Fennell et al. 2016; Ferjan Ramírez et al. 2017; Flege and Eefting 1986; Gonzales et al. 2019; Grosjean 1996; Luk and Bialystok 2013; Smith et al. 2021; Soto et al. 2019). Speech intelligibility occurs within the context of spoken language. Learning a second language (L2) occurs within the context of different learning processes than for monolinguals. Aspects of language that do not immediately or easily transfer from the vernacular first language (L1) to the second language (L2) are referred to as language interference (C.A. Brown 1998; Cummins 2009; Markov et al. 2022). Knowledge of common word sequences (lexical and collocational knowledge) for L2 likely develops across the lifespan; while, knowledge of grammar structures (morphosyntax) and individual speech sounds (phonetics) seem to have sensitive periods between six years and mid-adolescence, respectively (Granena and Long 2012). These developmental learning differences (i.e., in lexical word knowledge, morphosyntax, and phonetics) between L1 and L2 can result in perception and production differences in L2 (Anthony et al. 2009; Birdsong 2018; Fennell et al. 2016; Ferjan Ramírez et al. 2017; Flege and Eefting 1986; Granena and Long 2012; Grosjean and Miller 1994; Li 1996; Li and Yip 1998; Luk and Bialystok 2013; Smith et al. 2021). Native listeners perceive L2 production differences as a foreign accent. For example, the Spanish tap /r/ is not present in English and is often difficult for English speakers to perceive as a distinct sound and to acquire. Likewise, English "th" sounds, /ð/ and /θ/, do not occur in Western hemisphere Spanish and are difficult for Spanish speakers to perceive as distinct sounds and also to acquire (Yavaş and Goldstein 2018). In such cases, Goldstein and Bunta (2011) explain that a speaker will likely substitute these L2 sounds with the closest available approximation from L1 (e.g., for a Spanish speaker, /ðɪs/->/dɪs/). Both positive and negative cross-linguistic influences may be observed in both simultaneous and sequential bilinguals (Bialystok et al. 2009; Fricke et al. 2019; Navarro-Torres et al. 2021).

Birdsong (2018) stated, "It is important to emphasize that, despite bilingualism effects, there are late L2 learners who resemble native monolinguals with respect to targeted aspects of the L2 (as opposed to bilinguals being indistinguishable from monolinguals in every measurable respect)" (p. 6). Bilingual speakers, fluent in their native language, can achieve native-like abilities in speech voice onset times (VOT), global pronunciation, morphology, and syntax in the second language (Birdsong 2018). McCarthy et al. (2014) found that while bilingual children showed less consistent use of English words than their monolingual counterparts between 46 and 57 months of age; by the time the children began primary school, this difference disappeared and both groups had comparable vocabulary sizes. Consequently, research has shown that sequential bilingual learners are similar to

monolingual learners in terms of overall language skill acquisition (Castilla et al. 2009) and that neither simultaneous nor sequential bilingualism causes speech-language disorders (Korkman et al. 2012; Marinis and Chondrogianni 2011).

Other aspects of bilingualism include when L2 was learned (i.e., early vs. late learners) and how L2 was learned (i.e., simultaneous and sequential bilingual speakers). Research indicates that all early/late and simultaneous/sequential bilinguals are capable of native-like proficiency, receptively and expressively (Bak et al. 2014; Birdsong 2018; MacLeod and Stoel-Gammon 2005; McCarthy et al. 2014). Thus, indicating that any second language attained, generally speaking, is not always inferior to first language abilities. However, bilinguals and monolinguals are not identical in how they process and produce language. Kupisch et al. (2014) also assert that native-like proficiency is most common in bilinguals who use L2 on a daily basis over a sustained period of time. Herschensohn (2022) stated that both simultaneous and sequential learners are primarily influenced by internal and external factors affecting L2 acquisition.

Learning is substantially influenced by other factors including: (a) sensitive periods for language learning; (b) age of acquisition; (c) the amount of time spent using a language; and (d) the phonetic (sound) features of a language (Brice et al. 2021; Castilla et al. 2009; Kupisch et al. 2014; Herschensohn 2022; MacLeod and Stoel-Gammon 2005).

### 1.2. Second Language Proficiency

Two estimates of length of exposure to L2 are age of acquisition (AoA) and length of residence (LoR). Age of acquisition (AoA) refers to the age at which L2 is first encountered. An individual may arrive in the United States at three years of age, yet not be fully exposed to English until five years of age when they enter school. Hence, the age of arrival would not serve as an adequate indicator in this situation. Therefore, the age of acquisition would be a more accurate estimate of the length of exposure. LoR refers to the amount of time spent in contact with L2. Age of acquisition and LoR are used to make distinctions regarding the amount of L2 language exposure (i.e., early, middle, and late bilingualism). While estimates vary somewhat within the literature, Brice et al. (2013) defined early bilingualism as L2 onset between birth and 8 years of age, middle bilingualism between 9 and 15 years of age, and late bilingualism after 16 years of age. The study of speech perception and production through the lens of AoA and LoR can offer observable data on how individuals from the three bilingual age groups compare to monolinguals.

Granena and Long (2012) offer a detailed picture of how both AoA and LoR affect different aspects of language learning. For example, a battery of language mastery tests completed by those who acquired L2 during the upper limit of the sensitive period for phonetic skills (12 years of age) revealed a pattern of much greater proficiency with morphosyntax than with producing and identifying individual speech sounds (Granena and Long 2012). Thus, length of time speaking L2 (LoR) may have more of an impact on the ease with which an individual can converse in L2; while early exposure (AoA) may have more of an impact on how individual speech sounds are processed (Granena and Long 2012).

The findings of De Carli et al. (2015), corroborate those of Granena and Long (2012), who found that consistent and sustained practice of L2 was more important than AoA for achievement of high proficiency in conversational L2. Together, these findings indicate a possibility that, although bilinguals can become highly proficient in the recognition and use of L2 word sequences and grammar structures, there may be fundamental differences from monolingual/native speakers in terms of how phonotactic elements of speech are processed during speech perception. For example, Sebastián-Gallés et al. (2005) employed a gating task to examine how well Spanish/Catalan bilinguals recognized non-word phonemic contrasts. All stimuli consisted of disyllabic non-words with phonemic contrasts that were common to Catalan but nonexistent in Spanish. They found that participants with Spanish as their L1 required larger portions of the stimuli to make accurate identifications when

compared to participants with Catalan as their L1. Sebastián-Gallés et al. (2005) also found that the Spanish L1 speakers failed to differentiate /e/-/ɛ/ vowel contrasts.

These findings suggest that L1 impacts the perception of non-native phonemic contrasts, even when exposure to the second language is early and extensive. For example, a monolingual English speaker may not be able to perceive a Spanish tap or flap /r/ and perceive it as a /d/ sound instead (Brice et al. 2009). In a gating study with 30 monolingual and 30 Spanish-English speakers (early bilinguals, 0–8 years of age; middle bilinguals, 9–15 years of age, and late bilinguals, 16 plus years of age), Brice and Brice (2008) found that all the adult bilingual speakers identified words faster with initial voiced consonants than initial voiceless consonants. In addition, CV words that contained tense vowels were identified faster than words with lax vowels in English. Spanish does not have lax vowels; therefore, the perception of English-lax vowels will be affected (Kondaurova and Francis 2010; Smith and Hayes-Harb 2016). Consequently, sounds that are more common in L1 (e.g., tense vowels) can have lasting effects on L2 perception (e.g., lax vowels and slower speech perception) even into adulthood.

*1.3. Code-Switching and Code-Mixing*

It is common for bilingual speakers to use more than one language in their daily interactions. This use of both languages may manifest as code-switching or code-mixing. Other terms are used to describe this bilingual phenomenon (e.g., language alternations, use of translanguages, inter-sentential alterations, and intra-sentential alterations); however, we will use the terms code-switching and code-mixing. Code-switching refers to the use of both languages within a single discourse (Li 1996; Li and Yip 1998), usually across sentence boundaries. For example, a German-English bilingual speaker might say, "Hello there! *Wie geht's*? (How are you?)". Code mixing is the use of two languages within a single sentence (Martin et al. 2003). For example, "I like that *kleid* (dress)", or "*Andale pues* (okay, swell), and do come again, mm?" (Gumperz 1982). Gumperz (1982) offers another example in Spanish, taken from conversation: "She doesn't speak English, so, *dice que la regañan: 'Si se les va olvidar el idioma a las criaturas'* (she says that they would scold her: 'the children are surely going to forget their language')" (p. 76). While code-mixing is more syntactically complex than code-switching, both are common behaviors among bilingual speakers (Brice and Brice 2008; Brice et al. 2021; Grosjean 1996, 2001; Grosjean and Miller 1994; Heredia and Altarriba 2001; Li 1996; Li and Yip 1998). Most bilinguals voluntarily engage in purposeful code-mixing in natural conversation (Gollan and Ferreira 2009), and researchers have found that bilinguals readily use the speech sound inventories of both L1 and L2 when code-mixing (Genesee 2015; Grosjean 1996, 2001; Grosjean and Miller 1994). Code-mixing is not observed to be more taxing than choosing one language over the other and may in fact be easier than restricting speech to a single language (Gollan and Ferreira 2009), even though bilinguals may utilize more cognitive resources to choose which language to use when code-mixing as opposed to when speaking in a single language (Flege 1995; Li 1996). Most research examining code-mixing has focused on the ability of bilingual individuals to produce speech sounds in L1 and L2 (Genesee 2015; Gollan and Ferreira 2009; Grosjean 1996, 2001; Grosjean and Miller 1994; Heredia and Altarriba 2001; Piccinini and Arvaniti 2015; Thornburgh and Ryalls 1998). The limited literature on bilinguals has typically addressed speech production and focused significantly less on speech perception (López 2012; Sebastián-Gallés et al. 2005; Edwards and Zampini 2008). García-Sierra et al. (2012) stated, "However, little is known about the way speech perception in bilinguals is affected by the language they are using at the moment" (p. 194).

Fewer studies have focused on how code-mixing might affect speech perception in bilingual individuals, particularly in relation to monolingual individuals (Brice et al. 2013). The issue of English language proficiency among Spanish-English bilingual speakers cannot be made unless they are compared to monolingual speakers. Hence, many studies have

contrasted bilingual and monolingual populations (Boudelaa 2018; Delattre 1965; Ferjan Ramírez et al. 2017; Goldstein and Bunta 2011; García-Sierra et al. 2012; Jasinska and Petitto 2013; Marinis and Chondrogianni 2011). Brice and Brice (2008) also showed that bilingual speakers identified voiced initial consonant sounds quicker than initial voiceless consonants (e.g., /b/ vs. /p/). Brice et al. (2013) found that children and adult bilinguals were faster in recognizing voiceless English consonants than Spanish-voiceless consonants. The inclusion of children in this study may have altered perception patterns from those observed among strictly adults in the earlier Brice and Brice (2008) study.

The phonetic frequency of such sounds may have influenced the results of Brice et al. (2013). To give an example, high-frequency sounds are those that occur often in a language and are thus more easily recognizable (Metsala 1997). Thus, it is possible that the voiceless consonants had a high frequency of occurrence and were more easily recognized by participants (Brice et al. 2013; Geiss et al. 2022; Keating 1980). Additionally, the Spanish speakers in the Brice et al. (2013) study were all early bilinguals and may have had more exposure to English than to Spanish; whereas, the participants in the Brice and Brice (2008) study consisted of early, middle, and late bilinguals. Hence, bilinguals may change their perception of speech sounds as a result of their first English exposure and length of English exposure. Consequently, it is important to delineate AoA carefully.

In contrast, monolinguals need only deal with the functional load (i.e., the importance of certain features that assist in making language distinctions), phonetic frequencies, and word frequencies in one language. Functional load is not to be confused with cognitive load. Whereas cognitive load refers to mental effort (Sweller 1988), functional load refers to the speech sound's distinctiveness in the utterance context (A. Brown 1988). A phonetic feature with a high functional load is one that contributes greatly to making a word understandable. One method of investigating the way in which high-frequency sounds, functional loads, and cognitive loads contribute to an individual's perception of speech is through the use of gating.

*1.4. Gating*

Developed by Grosjean (1988), gating is when an auditory stimulus is presented to a listener in equal and increasing segments of time. The sound segments that a listener hears are referred to as "gates." The stimuli may range from single words to longer phrases and/or sentences. This study is concerned with gating single words such as in previous gating studies (Brice et al. 2021; Li 1996; Sebastián-Gallés et al. 2005). Gating is useful because it quantifies the extent of phonetic information that a listener needs in order to recognize and identify words (Li 1996).

Gating involves parsing an auditory stimulus presented to a listener into segments. Gates are typically presented in segments of 50–70 ms (Boudelaa 2018; Brice et al. 2013; Grosjean 1996; Li 1996; Mainela-Arnold et al. 2008). In this scenario, the listener first hears 70 ms of a word, then 140 ms of the word, and so on until either the listener makes a correct identification of the word or the end of the trial is reached. The outcome measures of interest are labeled the "isolation point" and the "recognition point" (Grosjean 1996). The isolation point is defined as the portion of the stimulus (e.g., 4 of 9 gates) needed for participants to make a correct identification of the target word. The recognition point in this study also refers to the portion of the stimulus needed for participants to recognize the word; however, the participants must give two consecutive correct identifications with 100% certainty. This determines the accuracy of their identification.

*1.5. Neuroimaging Investigations of Bilingual Language Perception and Production*

Monolingual and bilingual speech and language processing seems to require both comparable and dissimilar brain areas. Abutalebi et al. (2001) presented evidence that, in some instances, second language (L2) processing occurs in the same dedicated language areas as first language (L1). Questions have also been raised regarding possible bilingual-monolingual differences in the recruitment of other brain areas, such as the prefrontal

cortex, for language processing, particularly when bilingual individuals switch between multiple languages as with code-mixing or code-switching (Abutalebi et al. 2001). In a later review of fMRI and PET studies, Abutalebi (2007) explored evidence for such differences, finding that many studies supported the notion of increased brain activity in areas also associated with L1 language processing when bilinguals engaged in both code-switching and code-mixing. There is also a wealth of evidence for the recruitment of additional areas by bilingual language users when engaging in code-mixing and code-switching; specifically, the left prefrontal cortex (Brodmann areas 8, 9, 10, 11, 12, 13, 44, 45, 46, and 47), anterior cingulate cortex (ACC; Brodmann areas 24, 32, 33), and basal ganglia (Abutalebi 2007; Hernandez et al. 2000; Jasinska and Petitto 2013; Kovelman et al. 2008). Some researchers have begun investigating bilingualism and brain localization of function utilizing imaging technology, particularly fNIRS technology (Jasinska and Petitto 2013; Kovelman et al. 2008, 2014; Zinszer et al. 2015). Many researchers now do not believe Broca's area to be restricted in function to only speech production, nor Wernicke's area only to speech comprehension; it is more likely that all of the aforementioned language centers of the brain are involved both in comprehension and production of speech (Fadiga et al. 2009; Hagoort 2014). Hagoort (2014) provides an overview of an emerging, dynamic view of speech perception and production, in which speech production and comprehension act as shared networks among frontal, temporal, and parietal regions. It is also likely that more areas than just these dedicated speech centers are used for language. For example, Hagoort (2014) states that memory processes have been implicated in language processing, as individuals access a mental lexicon of speech sounds and whole words both when listening and when speaking.

Exploring differences not only between monolinguals and bilinguals but among bilinguals themselves, Hernandez (2009) conducted an investigation into differences in levels of neural activation between low-proficiency and high-proficiency bilinguals. When given a picture-naming task, bilinguals showed increased dorso-lateral prefrontal cortex (DLPFC; Brodmann areas 9 and 46) activation when switching between languages as opposed to naming pictures in a single language, regardless of the person's language proficiency level (i.e., low or high). Activation was also noted in brain areas devoted to the hippocampus (i.e., memory) and amygdala (i.e., somatosensory processing) in code-mixed language conditions and single-language conditions (Hernandez 2009). Hernandez speculates that somatosensory processing is involved due to emotions and cognition salient to the words in each language for the participants. The superior parietal lobe is also involved in somatosensory processing, attention, and visual-spatial perception (Wilkinson 1992). It was unclear, however, if the somatosensory processing activation is unique to bilinguals or if it is shared with monolinguals.

Although Hernandez's (2009) work addressed only early and late bilinguals, evidence from Archila-Suerte et al. (2015) supports the idea that neural processing both overlaps and differs between bilinguals and monolinguals and between early and late bilinguals. In an fMRI study of brain activity during both speech production and speech perception tasks, it was found that monolinguals, early bilinguals (those with an AoA of less than 9 years old), and middle bilinguals (those with an AoA of more than 10 years old) performed similarly in terms of results for speech production and perception in L2 (L1 for monolinguals).

It should be noted that the neural processing of speech sounds differed. Early bilinguals showed greater engagement of prefrontal regions involved in working memory compared to monolinguals, while middle bilinguals showed greater activation in the inferior parietal lobule compared to both early bilinguals and monolinguals (Archila-Suerte et al. 2015).

Similarly, Perani et al. (1998) found evidence for differences in activation in both left and right temporal and hippocampal regions between high and low proficiency groups in a PET investigation of performance on a task involving comprehension of an entire story. Low-proficiency bilinguals showed lower activation than high-proficiency bilinguals, though both bilingual groups displayed greater activity (more blood flow) located in areas

also associated with L1 (Perani et al. 1998). Together, this evidence suggests that although general localization of neural speech processing may be common between monolingual and bilingual groups; levels of activation and/or diverse neural regions may also differ. This supports the notion that perception and underlying language comprehension can be comparable yet also vary between bilingual and monolingual individuals.

*1.6. Functional near Infrared Spectroscopy (fNIRS)*

Functional near-infrared spectroscopy (fNIRS), a technique that uses infrared light to examine hemodynamic response in a shallow brain depth (approximately 3 cm), is an emerging method of investigating levels of neural activation. The regions of the brain that can be examined via fNIRS include the lateral prefrontal cortex (LPFC) and the medial prefrontal cortex (MPFC). Since a delay occurs between response to a stimulus and peak oxygenation readout on an fNIRS device (Tak and Ye 2014), most fNIRS researchers (Kovelman et al. 2008; Minagawa-Kawai et al. 2007; Zinszer et al. 2015) have employed block designs to capture data. These designs measure peak oxygenation levels during blocks of time starting approximately 5 s after the initial presentation of target stimuli and are measured against a baseline of oxygenation data.

In review, the focus of this study is based on the following principles:

1.  There is limited research and literature on bilingual speech perception (López 2012; Sebastián-Gallés et al. 2005; Edwards and Zampini 2008);
2.  Most bilingual speakers voluntarily engage in code-mixing in conversation (Gollan and Ferreira 2009) and bilinguals use speech sound inventories from both their L1 and L2 when code-mixing (Genesee 2015; Grosjean 1996, 2001; Grosjean and Miller 1994);
3.  Exploring the differences between speech sound inventories of monolinguals and bilinguals has been limited (Hernandez 2009);
4.  Code-mixing language offers the prospect of studying L2 speech perception while both languages are being assessed (López 2012); and
5.  Researchers have begun using fNIRS imaging technology to investigate bilingualism and brain localization function (Jasinska and Petitto 2013; Kovelman et al. 2008, 2014; Zinszer et al. 2015).

*1.7. Purpose*

Questions still remain regarding the patterns of language activation within a bilingual brain (e.g., does it occur in the same dedicated area for L1 for early bilinguals compared to monolingual adults, in different areas, or in both?) (Abutalebi 2007; Abutalebi et al. 2001). Bilingual listeners may be able to identify words and sounds unique to each language better than words and sounds common to both languages. The speech learning model (SLM) states that phonologic cues tend to be more recognizable when they are specific only to one language than when the cues are common to both languages (Flege 1995). Features such as voiced/voiceless consonants and vowel tenseness (lax or tense vowels) may serve to narrow word and sound choices between and within languages. Features of neural activation will also be investigated to determine language-specific responses within specified areas of the prefrontal cortex, specifically the lateral prefrontal cortex (LPFC), and the medial prefrontal cortex (MPFC) in both the left and right hemispheres. These areas are accessible and measurable with the use of fNIRS. Consequently, the purpose of this study is to investigate speech perception and word identification in both monolingual and bilingual neurotypical populations via neural activation monitoring utilizing functional near-infrared spectroscopy (fNIRS).

*1.8. Research Questions and Hypotheses*

Research questions one and two both utilized isolation points, recognition points, and gating change scores as the dependent variables. Whereas, research questions three and four employed blood oxygenation as the dependent variable. The research questions and hypotheses were as follows:

1.  Do bilingual speakers identify sounds (phonotactics) better in one language better than the other? Are bilingual speakers comparable in their L2 to native English speakers? It is expected that there will be differences in English and Spanish phonotactics between bilingual (Spanish-English) speakers and monolingual (English) speakers for isolation points, recognition points, and gating change scores;
2.  Do bilingual speakers identify words and languages quicker in code-mixed vs. non-mixed sentences? Do English speakers identify words quicker in non-mixed sentences than in code-mixed sentences? It is expected that there will be differences for words and languages across the four code-mixed and non-mixed language conditions (i.e., English-English (non-mixed); English-Spanish (code-mixed); Spanish-Spanish (non-mixed); Spanish-English (code-mixed)) between bilingual (Spanish-English) speakers and monolingual (English) speakers for isolation points, recognition points and gating change scores;
3.  Do bilingual and monolingual speakers share the same, similar, or different neural activation areas when identifying words and languages in code-mixed vs. non-mixed sentences? It is expected that there will be differences in neural activation and languages between bilingual (Spanish-English) speakers and monolingual (English) speakers for blood oxygenation levels; and
4.  Do bilingual and monolingual speakers share the same, similar, or different neural activation areas when identifying sounds (phonotactics) in both languages? It is expected that there will be differences in neural activation and phonotactics across four code-mixed and non-mixed language conditions between bilingual (Spanish-English) speakers and monolingual (English) speakers for blood oxygenation levels.

## 2. Materials and Methods

All methods were approved by the Institutional Review Board (IRB) at the university where this research was conducted (IRB#: Pro00010922). All participants gave informed written consent prior to participation in the study.

*2.1. Participants*

Individuals who spoke only English and those who spoke both Spanish as their L1 and consequently English (early AoA, 3–8 years) served as participants. Individuals with speech-language and/or hearing disabilities were excluded from this study based on self-identification. A total of 20 participants (10 monolingual, 10 bilingual) remained in the present study after exclusions based on age (18–46; mode: 20), language status, and lack of speech-language and/or hearing disability. All bilingual participants were proficient in both spoken languages (i.e., Advanced to Native-like proficiency as measured by the *ISPLR*). Age of onset for L2 occurred prior to eight years of age among all of the bilingual participants.

*2.2. Materials*

Language proficiency rating. The *International Second Language Proficiency Rating* tool (*ISLPR*; Wylie and Ingram 2010) was administered by one of the researchers who was highly proficient in both Spanish and English. A proficiency level of three or higher (Advanced to Native-like) from the *International Second Language Proficiency Rating* tool (*ISLPR*; Wylie and Ingram 2010) was used to select the bilingual participants. Bilingual participants were interviewed and rated on their Spanish, English, and code-mixing abilities. Monolingual participants were interviewed only in English. In addition, all participants self-rated their language abilities. Consequently, the participants were formally classified as bilingual or

monolingual. Participants with an oral proficiency rating of three (i.e., deemed proficient) or higher in both English and Spanish were classified as bilingual; those with a proficiency score below the cutoff (as set by Wylie and Ingram 2010) in Spanish were classified as monolingual speakers of North American English. None of the participants' self-ratings differed from that of the examiner by more than one point on the *International Second Language Proficiency Rating* tool (*ISLPR*; Wylie and Ingram 2010) (i.e., 1–5 on a five-point scale with one being the lowest score and five the highest). Monolingual participants mostly indicated some exposure to Spanish through living in a state with a high Spanish-speaking population or having taken Spanish courses in their academic careers. However, none of the monolingual participants indicated any proficiency levels beyond simple words or phrases, essentially level one on the *International Second Language Proficiency Ratings* (*ISLPR*) (Wylie and Ingram 2010).

Equipment and software. Neuroimaging data were collected using a Biopac fNIRS recording box and the fNIRS Software Version 3.4 (Biopac 1997) and 16-channel forehead-mounted sensor band (see Figure 1 for optode placement). The sensor band contained four infrared light sources and ten detectors. The gating task and language stimuli were presented auditorily using a modified version of Grosjean's (1996) gating paradigm, presented to participants in E-prime 2.0 (Psychology Software Tools, Pittsburgh, PA, USA) via a Dell desktop computer (displaying 12-point font). Participants were seated approximately 1–2 feet away from the screen. The gated segments were heard through Sennheiser over-the-ear HD-201S headphones. All participants reported no difficulty seeing and/or hearing the stimuli. See Figure 1 for optode placement and illustration of the fNIRS band and equipment. Figure 2 provides an image of an individual's single response (i.e., one gate presentation) across the 16 fNIRS diodes during the experiment. Odd-numbered fNIRS diodes are seen along the top; while even-numbered fNIRS diodes are seen along the bottom. fNIRS Diode 1 is seen in the upper corner (left Lateral Prefontal Cortex, LPFC); while, Diode 2 (left Lateral Prefrontal Cortex, LPFC) is seen below. Diode 16 (right Lateral Prefrontal Cortex, LPFC) is seen in the lower right corner; while, Diode 15 (right Lateral Prefrontal Cortex, LPFC) is seen above Diode 16.

Language stimuli. In creating the stimuli, the recorded sentence and word presentations were initially counterbalanced to achieve the following four language conditions: (a) English sentence—English word (EE, non-mixed); (b) English sentence—Spanish word (ES, code-mixed); (c) Spanish sentence—English word (SE, code-mixed); and (d) Spanish sentence—Spanish word (SS, non-mixed). The words were also classified according to the following four phonotactic conditions: (a) voiced consonant-consonant-vowel (CCV+); (b) voiceless consonant-consonant-vowel, (CCV-); (c) tense consonant-vowel (CV tense); and (d) lax consonant-vowel (CV lax). The CV lax condition contained only English words due to the low occurrence in Spanish (Delattre 1965).

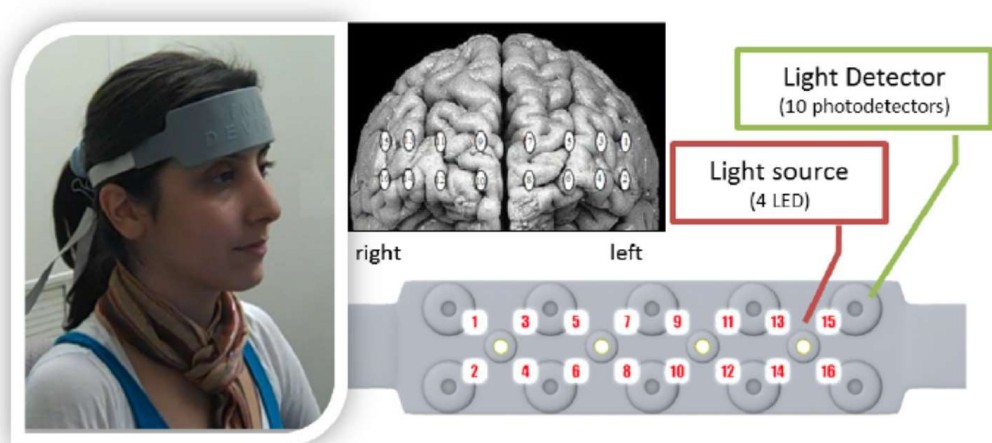

**Figure 1.** Illustration of fNIRS band and optode placements.

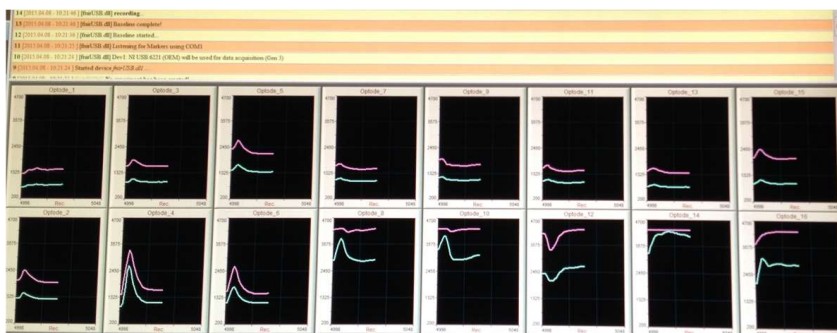

**Figure 2.** fNIRS processing across 16 optodes.

Sentence-word pairs were arranged in 60 trials. All target words were nouns that occurred at the end of a carrier phrase. Carrier phrases were generic to only indicate top-down sentence contextual information in that the target word was a "noun". The list of phrases and stimuli words is provided in Appendix A. All sentence-word combinations were spoken by a fluent, female Spanish-English speaker. The speaker used careful articulation and a neutral, Costa Rican accent in Spanish and a general American English (GAE) accent in English. The speaker displayed no discernable Spanish accents when speaking English or English accents when speaking Spanish. The sentence-word-audio stimuli were used in prior published studies (Brice and Brice 2008; Brice et al. 2013, 2021). Forty of the 60 stimuli were code-mixed, containing Spanish and English words; code-mixed stimuli began with a sentence in one language and ended with a target word in the other language (e.g., "I see a *brillo*" (I see a brightness), "Quiero un *cookie*" (I want a cookie)). Nine stimuli were presented entirely in Spanish and 11 were presented entirely in English. Single-language sentences (i.e., English-English or Spanish-Spanish) were randomly interspersed among the code-mixed sentences to prevent participants from predicting patterns of code-mixed target stimuli. All test items were presented in the same sequence for all participants (i.e., the stimuli were pre-recorded and used from previous studies, thus allowing comparison to the previous studies). The above construction resulted in four language conditions: English sentence—English word (EE), English sentence—Spanish word (ES), Spanish sentence—English word (SE), and Spanish sentence—Spanish word (SS).

*2.3. Procedure*

Each participant provided written consent as per the university Institutional Review Board (IRB) procedures. Following this, the experimenter discussed the experiment and held a conversation with the participants to determine their level of English and Spanish language proficiency (as measured by the examiner using the *ISPLR*). Participants were seated at the computer and fitted with the fNIRS band. Fitting both the fNIRS band and Sennheiser headphones was a two-three minute procedure (See Figure 1). In order to facilitate fNIRS recording, light levels were dimmed so that only the emergency light remained on in the room. The E-prime gating task was initiated. Participants sat at a computer screen with a keyboard. The experimenter was seated in the same room and evaluated the correctness of the verbal response via mouse click. The participant's computer was attached to two monitors set back-to-back. Participants were seated at one monitor with the experimenter seated directly across at another monitor judging the correctness of responses. Participants were unable to see the experimenter evaluating and recording responses.

Participants were presented with sentences auditorily over the Sennheiser headphones and asked to identify the last word in the sentence. Participants were presented with five practice items followed by 60 experimental items (40 code-mixed; 9 Spanish; 11 English). Example sentences participants heard included general carrier phrases such as, "Quiero un *plate*" (I want a plate) or "I saw the *bruja*" (I saw the witch). The number of gates per word ranged from 4 to 14. Each gate was approximately 65–70 milliseconds in duration.

Participants first heard a carrier sentence plus the first gate. Following this, each subsequent gate was added to the duration of the prior gates until the full word was identified. The carrier sentence was not presented in between gates. Participants followed on-screen questions after each stimulus presentation and reported via a keyboard click whether or not they recognized the word. If not, the next gate was presented. If they recognized the word (i.e., isolation point), they were prompted on-screen to say the word out loud. The responses were immediately noted by the experimenter, i.e., not recorded. The stimulus trial ended when the participant gave two consecutive correct identifications with 100% confidence (i.e., the recognition point) or when all gates had been presented. It was rare that participants reached the end of the stimuli without a recognition point. Each experiment lasted approximately one hour in duration. Participants wore the fNIRS headband throughout the duration of the experiment. None of the participants reported discomfort during the experiment.

### 2.4. Non-Parametric Statistics

Mann–Whitney U tests (Mann and Whitney 1947; Nachar 2008) were conducted for all phonotactic (sound) conditions. The Mann–Whitney U test does not assume any properties regarding the distribution of the underlying variables in the analysis or sample size (Sullivan 2022). Alpha in this study was set at 0.001 after Bonferroni corrections. In addition, the two basic assumptions of independent samples and homogeneity of variance were met (Cohen 1988). The samples were independent (i.e., as each participant was unrelated to the others in the sample). In addition, Levene's test indicated non-significant results for all phonotactic, language measures, and fNIRS results by group; hence, homogeneity of variance was assumed (Tomkins and Hall 2006).

This study used both cross-sectional and repeated measures as is typically done with gating studies (Brice and Brice 2008; Brice et al. 2013, 2021; Grosjean 1996; Li 1996; Guo et al. 2013). Please note that an increase in the number of comparisons conducted (i.e., each sample should contain five or more measures) increases the power of the test (Cardone 2010; Mumby 2002). The repeated measures data contained 806 data cells by 20 participants (i.e., 16,120 separate measures). Consequently, the use of non-parametric statistics reported seemed justified.

### 2.5. Language Conditions Data Analysis

Phonotactic isolation point data were examined using a two-way repeated measures ANOVA with language group (bilingual, monolingual) as the between-subjects variable and sentence-word condition (English-English, Spanish-Spanish, English-Spanish, and Spanish-English) as the within-subjects variable. The same strategy was then repeated for recognition point data (the proportion of the word, as measured in 70 ms gates, needed to elicit two correct verbal identifications with 100% accuracy), and for examining the differences between isolation and recognition points. Because the stimulus items had differing numbers of gated segments from 4 to 14, the gating data were transformed into percentages in order to compare across the various word lengths of the stimulus items. Mean values for isolation point and recognition point are to be interpreted as follows. Scores can range from null to one and represent portions of the full word. For example, a value of 1.00 indicates 100% of gated word segments, and a value of 0.5 represents 50% of gated word segments. Mean ranks and standardized test statistics for Mann–Whitney U one-tailed tests are reported.

### 3. Results

#### 3.1. Language Conditions and Phonotactics

The phonotactic isolation point data was examined using Mann–Whitney U to compare the language groups (bilingual, monolingual). Four analyses were conducted across the four language conditions: (a) English-English (non-mixed); (b) Spanish-Spanish (non-mixed); (c) English-Spanish (code-mixed); and (d) Spanish-English (code-mixed). The same

strategy was then repeated twice: for recognition point data and change score data. The recognition point was the proportion of the word, as measured in 70 ms gates, needed to elicit two consecutive and correct verbal identifications with 100% accuracy. Change score data consisted of differences between isolation and recognition points. Change scores indicate the length of time between identifying the sound (i.e., identification point) and making a final decision (i.e., recognition point). Large change scores indicated that participants took a longer time to make decisions and vice versa, shorter change scores indicated quicker decision-making. Hence, three data points are provided for the gated phonotactic data: (a) Isolation point; (b) Recognition point; and (c) Change score.

### 3.2. Language Conditions-between Subject Differences

For the neuroimaging analysis, prior to Bonferroni correction, there was a significant difference between monolinguals and bilinguals for optodes 5 (left Medial Prefrontal Cortex, MPFC) (U (20) = 2.192, $p$ = 0.029), 11 (right Medial Prefrontal Cortex, MPFC) (U (20) = 2.192, $p$ = 0.028), and 14 (right Lateral Prefrontal Cortex. LPFC) (U (20) = 2.570, $p$ = 0.010) in the Spanish-English condition. Bilinguals exhibited higher blood oxygen levels than monolinguals. No other significant differences were found (see Table 1).

**Table 1.** fNIRS Optode by Language Condition Prior to Bonferroni Correction.

| | Language Status | | | |
|---|---|---|---|---|
| fNIRS Optode | English-English Probability | English-Spanish Probability | Spanish-Spanish Probability | Spanish-English Probability |
| 1 | 0.971 | 0.684 | 0.631 | 0.481 |
| 2 | 0.853 | 0.739 | 0.353 | 0.393 |
| 3 | 0.393 | 0.853 | 0.739 | 0.579 |
| 4 | 1.000 | 0.218 | 0.393 | 0.684 |
| 5 | 0.143 | 0.481 | 0.315 | 0.029 * |
| 6 | 1.000 | 0.971 | 0.529 | 0.052 |
| 7 | 0.063 | 0.853 | 0.105 | 0.393 |
| 8 | 0.190 | 0.631 | 0.353 | 0.165 |
| 9 | 0.190 | 0.739 | 0.436 | 0.315 |
| 10 | 0.218 | 0.684 | 0.280 | 0.089 |
| 11 | 0.247 | 0.853 | 0.247 | 0.029 * |
| 12 | 0.579 | 0.631 | 0.247 | 0.063 |
| 13 | 0.579 | 0.912 | 0.353 | 0.280 |
| 14 | 0.280 | 0.529 | 0.075 | 0.009 * |
| 15 | 0.684 | 0.315 | 0.684 | 0.796 |
| 16 | 0.631 | 0.436 | 0.529 | 0.190 |

* $\alpha \leq 0.01$.

After Bonferroni corrections, no significant differences were found for isolation, recognition, and change scores between monolinguals and bilinguals for all four word and language conditions. For recognition scores, bilinguals were quicker at identifying words in the Spanish-Spanish (non-mixed) (i.e., bilinguals 6.67 ms vs. monolinguals 13.00 ms) and English-Spanish (code-mixed) (i.e., bilinguals 7.11 ms vs. monolinguals 12.60 ms) word and language conditions. Please see Table 2.

**Table 2.** Differences in Isolation, Recognition, and Change Across Language Conditions for Monolinguals and Bilinguals (* Prior to Bonferroni corrections). Monolingual and Bilingual values indicate Isolation and Recognition response times measured in milliseconds.

| | Language Status | | | |
| --- | --- | --- | --- | --- |
| **Isolation** | **Monolingual (n = 10)** | **Bilingual (n = 10)** | *Mann–Whitney U* | **Prob** |
| English-English | 10.20 | 10.80 | 0.227 | 0.853 |
| English-Spanish | 12.00 | 9.00 | −1.134 | 0.280 |
| Spanish-Spanish | 12.90 | 8.10 | −1.814 | 0.075 |
| Spanish-English | 11.20 | 9.80 | −0.429 | 0.631 |
| Recognition | | | | |
| English-English | 8.60 | 11.56 | 0.253 | 0.278 |
| English-Spanish | 12.60 | 7.11 | −2.124 | 0.035 * |
| Spanish-Spanish | 13.00 | 6.67 | −2.449 | 0.013 * |
| Spanish-English | 11.00 | 10.00 | −0.378 | 0.739 |
| Change | | | | |
| English-English | 10.30 | 9.67 | −2.45 | 0.842 |
| English-Spanish | 11.10 | 8.78 | −0.898 | 0.400 |
| Spanish-Spanish | 9.40 | 10.67 | 0.490 | 0.661 |
| Spanish-English | 9.50 | 11.50 | 0.756 | 0.481 |

* $\alpha \leq 0.01$.

### 3.3. Phonotactic Conditions-between Subject Differences

Mann–Whitney U tests were also conducted to compare bilinguals and monolinguals on phonotactic (sound) conditions. Items on the gating task were classified according to whether the gated word was voiced vs. voiceless and tense vs. lax. Both English and Spanish gated words were combined and grouped according to the language of the carrier sentence and phonotactic condition. No significant differences were found between subjects for isolation, recognition, and change scores (after Bonferroni corrections; $\alpha \leq 0.01$); however, it should be noted that bilinguals were quicker ($p = 0.035$) at isolating voiced words in English when compared to monolinguals. Bilinguals were also quicker ($p = 0.028$) at recognizing tense words in Spanish compared to monolinguals. No significant difference in change scores was observed. Please see Table 3.

**Table 3.** Differences in Isolation, Recognition, and Proportion Change Across Phonotactic Conditions (After Bonferroni corrections; * $\alpha \leq 0.001$). Monolingual and Bilingual values indicate Isolation and Recognition response times measured in milliseconds.

| | Language Status | | | |
| --- | --- | --- | --- | --- |
| **Phonotactic Condition** | **Monolingual (n = 10)** | **Bilingual (n = 10)** | *Mann–Whitney U* | **Prob *** |
| Isolation | | | | |
| English-Tense | 11.00 | 10.00 | 45.00 | 0.739 |
| Spanish-Tense | 11.90 | 9.10 | 36.00 | 0.315 |
| English-Lax | 11.15 | 9.85 | 43.50 | 0.631 |
| Spanish-Lax | 9.95 | 11.05 | 55.50 | 0.684 |
| English-Voiced | 13.30 | 7.70 | 22.00 | 0.035 |
| Spanish-Voiced | 12.90 | 8.10 | 26.00 | 0.075 |
| English-Voiceless | 11.20 | 9.80 | 43.00 | 0.631 |
| Spanish-Voiceless | 11.00 | 10.00 | 45.00 | 0.739 |
| Recognition | | | | |
| English-Tense | 11.50 | 8.33 | 30.00 | 0.243 |
| Spanish-Tense | 12.70 | 7.00 | 18.00 | 0.028 |
| English-Lax | 9.75 | 11.25 | 57.50 | 0.579 |
| Spanish-Lax | 10.60 | 9.33 | 39.00 | 0.661 |

**Table 3.** *Cont.*

| | Language Status | | | |
| --- | --- | --- | --- | --- |
| **Phonotactic Condition** | **Monolingual (n = 10)** | **Bilingual (n = 10)** | *Mann–Whitney U* | **Prob \*** |
| English-Voiced | 11.90 | 7.89 | 26.00 | 0.133 |
| Spanish-Voiced | 11.90 | 7.89 | 26.00 | 0.133 |
| English-Voiceless | 11.60 | 8.22 | 29.00 | 0.211 |
| Spanish-Voiceless | 9.70 | 10.33 | 48.00 | 0.842 |
| Change | | | | |
| English-Tense | 11.30 | 8.56 | 32.00 | 0.315 |
| Spanish-Tense | 11.65 | 8.17 | 28.50 | 0.182 |
| English-Lax | 9.15 | 11.85 | 63.50 | 0.315 |
| Spanish-Lax | 11.55 | 8.28 | 29.50 | 0.211 |
| English-Voiced | 9.00 | 11.11 | 55.00 | 0.447 |
| Spanish-Voiced | 9.10 | 11.00 | 54.00 | 0.497 |
| English-Voiceless | 11.15 | 8.72 | 33.50 | 0.356 |
| Spanish-Voiceless | 9.85 | 10.17 | 46.50 | 0.905 |

\* $\alpha \leq 0.001$.

### 3.4. Within-Group Differences

Friedman's ANOVAs (non-parametric tests for repeated measures) were conducted to compare each word and language condition within monolinguals and bilinguals for isolation, recognition, and change scores (after Bonferroni corrections; $\alpha \leq 0.01$) (See Table 3). The change score proportions shown in parentheses in Table 3 indicate the proportion of time between the isolation and recognition point scores (i.e., the proportion amount of time between initial identification of the word and ascertainment of the word). For monolinguals, there was a significant difference between English-English (non-mixed), English-Spanish (code-mixed), Spanish-Spanish (non-mixed), and English-Spanish (code-mixed) for isolation and recognition scores; with a trend towards significance for change scores. No significant differences were found for bilinguals. Refer to Table 4.

**Table 4.** Friedman's ANOVA Comparing Mean Rank for Proportion Isolation, Recognition, and Change Across Language Conditions (After Bonferroni corrections \* $\alpha \leq 0.001$). Values indicate Isolation and Recognition response times measured in milliseconds.

| | | Language Condition | | | | | |
| --- | --- | --- | --- | --- | --- | --- | --- |
| **Language Group** | **Point** | **English-English** | **English-Spanish** | **Spanish-Spanish** | **Spanish-English** | **Friedman** | **Prob** |
| Monolingual | Isolation | 1.10 (0.555) | 2.80 (0.658) | 3.80 (0.719) | 2.30 (0.633) | 22.680 | <0.001 * |
| | Recognition | 1.50 (0.793) | 3.60 (0.935) | 3.20 (0.922) | 1.70 (0.815) | 20.04 | <0.001 * |
| | Change | 2.90 (0.238) | 3.50 (0.281) | 1.80 (0.202) | 1.80 (0.183) | 12.84 | 0.005 ** |
| Bilingual | Isolation | 2.30 (0.564) | 2.30 (0.549) | 2.30 (0.575) | 3.10 (0.577) | 2.880 | 0.410 |
| | Recognition | 2.89 (0.846) | 2.44 (0.825) | 2.33 (0.825) | 2.33 (0.792) | 1.133 | 0.769 |
| | Change | 3.00 (0.260) | 3.11 (0.251) | 2.11 (0.216) | 1.78 (0.239) | 7.000 | 0.072 |

\* $\alpha \leq 0.001$; \*\* Trend toward significance $\alpha \leq 0.05$.

Results indicated no significant differences with $p \leq 0.001$ for all conditions; however, trends toward significance were found with $p \leq 0.05$. For isolation scores, monolinguals isolated words sooner in (a) the English-English (non-mixed) condition than the Spanish-English (code-mixed) condition ($p = 0.038$); (b) followed by the English-Spanish (code-mixed) condition ($p = 0.003$); and (c) then the Spanish-Spanish (non-mixed) condition ($p < 0.001$). There was no difference in the Spanish-English (code-mixed) and English-Spanish condition (code-mixed) ($p = 0.386$) or Spanish-Spanish (non-mixed) condition ($p = 0.083$). Monolinguals isolated words faster in the Spanish-English (code-mixed) condition than in the Spanish-Spanish condition (non-mixed) ($p = 0.009$). See Table 5.

**Table 5.** Spearman's Rank Correlations for Language Conditions (After Bonferroni corrections * $\alpha \leq 0.001$).

| Group | Condition | Point | Optode Number | Spearman's Rank Corr | Prob |
|---|---|---|---|---|---|
| Monolingual | English-English | Isolation | 11 (right MPFC) | 0.661 | 0.038 ** |
| | | Recognition | -- | -- | -- |
| | | Change | 14 (right LPFC) | −0.661 | 0.038 ** |
| | English-Spanish | Isolation | 12 (right MPFC)<br>14 (right, LPFC) | 0.636<br>0.709 | 0.048 **<br>0.022 ** |
| | | Recognition | 2 (left LPFC)<br>12 (right MPFC) | 0.694<br>0.673 | 0.025 **<br>0.033 ** |
| | | Change | -- | -- | -- |
| Bilingual | Spanish-Spanish | Isolation | 2 (left LPFC) | −0.636 | 0.048 ** |
| | | Recognition | -- | -- | -- |
| | | Change | 9 (right MPFC) | −0.683 | 0.042 ** |
| | Spanish-English | Isolation | -- | -- | -- |
| | | Recognition | 16 (right LPFC) | −0.661 | 0.038 ** |
| | | Change | -- | -- | -- |

* $\alpha \leq 0.001$; ** Trend toward significance $\alpha \leq 0.05$.

Spearman rank correlations (see Table 6) indicated the following: For recognition scores, monolinguals recognized words in (a) the English-English (non-mixed) condition faster than the Spanish-Spanish (non-mixed) condition ($p = 0.003$); and (b) then the English-Spanish (code-mixed) condition ($p < 0.001$); but not in (c) the Spanish-English (code-mixed) condition ($p = 0.729$). Monolinguals recognized words in the Spanish-English (code-mixed) condition sooner than Spanish-Spanish (non-mixed) ($p = 0.009$) and English-Spanish (code-mixed) ($p = 0.001$). There was no difference between the Spanish-Spanish (non-mixed) and English-Spanish (code-mixed) conditions ($p = 0.488$).

For change scores, Friedman's ANOVA in monolinguals indicated trends toward significance between isolation and recognition when comparing English-Spanish (code-mixed) (proportion = 0.281 ms) to both Spanish-Spanish (non-mixed) (proportion = 0.202 ms) and Spanish-English (code-mixed) (proportion = 0.183 ms), $p = 0.003$ (for both comparisons). The longest lag between isolation and recognition occurred for the English-Spanish (code-mixed) condition. The proportion change for English-English (non-mixed) was not significant ($p = 0.005$).

**Table 6.** Spearman's Rank Correlations for Phonotactic Conditions.

| Group | Condition | Point | Optode Number | Spearman's Rank Corr | Prob |
|---|---|---|---|---|---|
| Monolingual | English-Voiced | Isolation | 5 (left MPFC) | 0.830 | 0.003 * |
| | | | 7 (left MPFC) | 0.855 | 0.002 * |
| | | Change | 7 (left MPFC) | −0.745 | 0.013 |
| Bilingual | Spanish-Tense | Recognition | 2 (left LPFC) | −0.736 | 0.024 |
| | | | 5 (left MPFC) | −0.778 | 0.014 |
| | English-Tense | Change | 1 (left LPFC) | 0.667 | 0.050 |
| | | | 3 (left LPFC) | 0.773 | 0.025 |
| | Spanish-Tense | Change | 1 (left LPFC) | 0.700 | 0.036 |
| | | | 2 (left LPFC) | 0.717 | 0.030 |
| | | | 4 (left LPFC) | 0.867 | 0.002 * |
| | | | 6 (left MPFC) | 0.733 | 0.025 |
| | | | 14 (right MPC) | 0.667 | 0.050 |
| | English-Lax | Change | 5 (left MPFC) | −0.657 | 0.025 |
| | | | 9 (right MPFC) | −0.745 | 0.013 |
| | | | 11 (right MPFC) | −0.770 | 0.009 * |
| | | | 4 (left LPFC) | −0.794 | 0.006 * |
| | Spanish-Lax | Change | 5 (left MPFC) | −0.728 | 0.026 |

* $\alpha \leq 0.01$.

### 3.5. Neural Activation with fNIRS

Neuroimaging results included data from the fNIRS band measuring oxygenated blood levels across the frontal lobe. Oxygenation data for each optode (i.e., optical sensor) was detrended (i.e., removing a trend from the time series data) and processed using fNIRSoft (Biopac® 1997) (Biopic Systems, Inc., Irvine, CA, USA. Sourced directly from the manufacturer) Software Version 3.4. The dependent variable was the change in oxygenation levels compared to the baseline across all 16 optodes. Scores included both negative and positive values. Positive numbers indicated blood oxygenation rising above baseline levels; negative values indicated blood oxygenation dropping below baseline levels. The greater the number (either positive or negative), the greater the change from baseline or resting state.

In line with previous studies (Kovelman et al. 2008; Minagawa-Kawai et al. 2007; Zinszer et al. 2015), changes in blood-oxygenation levels over the course of each trial, from stimulus onset to identification of the word, were examined. Differences between bilingual and monolingual participants were examined for each of the 16 optodes using a series of Mann–Whitney U tests for each language condition: (a) English sentences—English words; (b) English sentences—Spanish words; (c) Spanish sentences—Spanish words; and (d) Spanish Sentences—English words. Phonotactics conditions included: (a) voiced vs. voiceless consonants; and (b) tense vs. lax vowels. Bonferroni corrections for the number of analyses conducted indicated a p-value of less than 0.001 would be considered significant.

Although there were no significant differences after Bonferroni correction, a review of the figures illustrates that activation was higher when the carrier sentence and gated word matched the speaker's L1 (i.e., English-English (non-mixed) for monolingual speakers and Spanish-Spanish (non-mixed) language conditions for the bilingual speakers). This difference was most evident for the English-speaking monolinguals when the carrier sentence was in their L1 (i.e., English). As shown in Figure 3, monolinguals exhibit higher activation than bilinguals for the English-English (non-mixed) condition. Figure 4 illustrates more overlap when the carrier sentence and gated word were code-mixed between English and Spanish than when complete monolingual utterances are presented (i.e., English-English; or Spanish-Spanish non-mixed language conditions). For example, Figure 4 shows slightly higher activation for monolinguals when the carrier sentence is in English and the gated word is in Spanish. In Figure 5, bilinguals exhibit slightly higher activation than

monolinguals in the Spanish-Spanish (non-mixed) condition. Figure 6 also showed more overlap when the carrier sentence and gated word were code-mixed between English and Spanish, i.e., carrier sentence in Spanish and the gated word in English. The differences in neural activation are shown in Figures 3–6. Please note that lower numbers indicate faster responses. The fNIRS diodes are numbered along the bottom on the X-axis (i.e., 1–16).

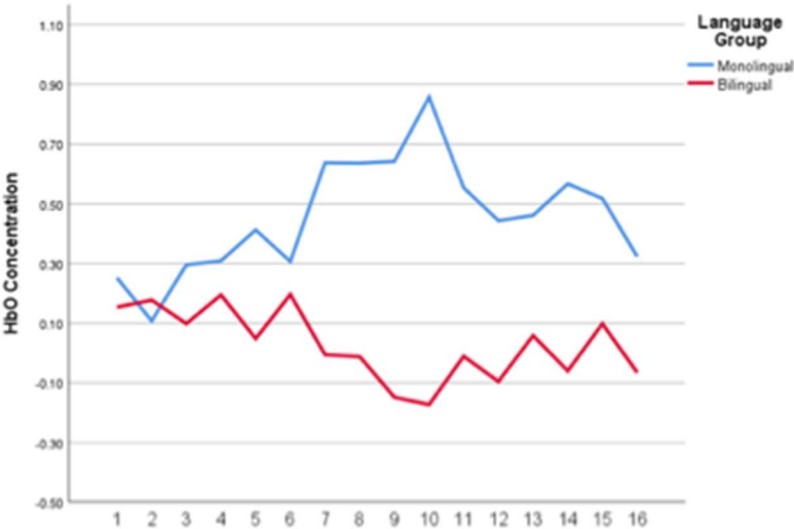

**Figure 3.** English-English Language Conditions Across 16 Optodes.

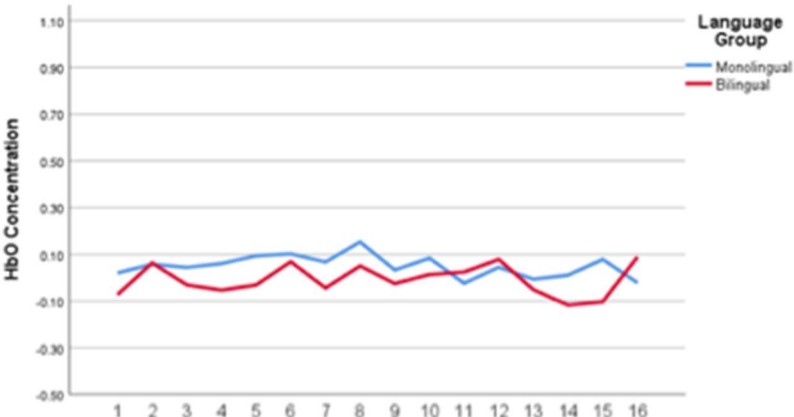

**Figure 4.** English-Spanish Language Conditions Across 16 Optodes.

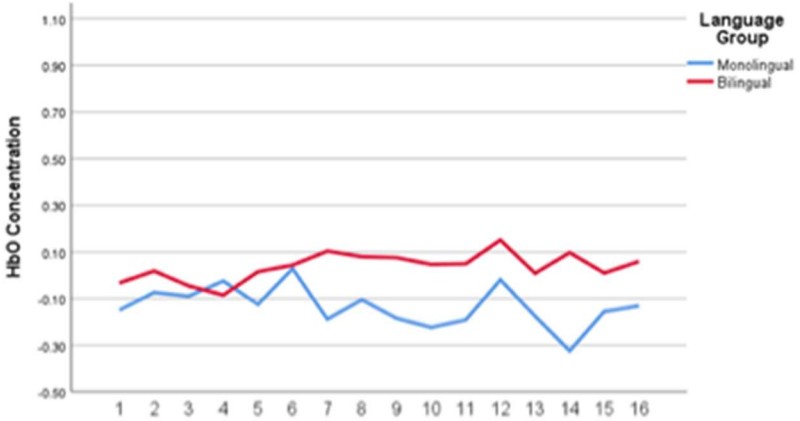

**Figure 5.** Spanish-Spanish Language Conditions Across 16 Optodes.

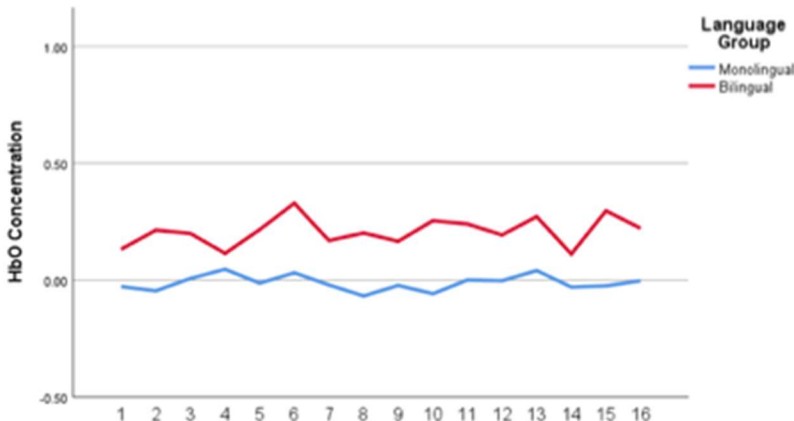

**Figure 6.** Spanish-English Language Conditions Across 16 Optodes.

### 3.6. Gating Task and Neural Activation in fNIRS

Spearman's rank correlations were computed for monolinguals and bilinguals under each language and phonotactic condition (after Bonferroni corrections; $\alpha \leq 0.01$). These correlations were conducted in order to assess the relationship between neural activation and word performance on the gating task. Spearman's rank correlations were computed between the proportion isolation/recognition/change and neural activation across the optodes for each language condition (English sentences—English words, English sentences—Spanish words, Spanish sentences—Spanish words, and Spanish Sentences—English words). No other significant differences were found. Please refer to Table 5.

When examining phonotactic conditions, both monolinguals and bilinguals exhibited significant relationships but only when the carrier sentence reflected their dominant language. The relationship was in opposite directions for monolinguals and bilinguals. For monolinguals, increased activation was related to longer isolation (EE and ES) and recognition (ES) points. This occurred bilaterally across widespread areas of the prefrontal cortex: Optodes 2 (left LPFC), 11 (right MPFC), 12 (right MPFC), and 14 (right LPFC). For bilinguals, increased activation was related to shorter isolation (SS) and recognition points (SE). This activation occurred in fewer areas bilaterally and overlapped somewhat with the areas activated for monolinguals: 2 (left LPCF) and 16 (right LPFC). Optodes 2 (left LPFC) and 16 (right LPFC) are the most lateral optodes and are located closest to homologous language areas in the left and right hemispheres. Optodes 11 (right MPFC), 12 (right MPFC), and 14 (right LPFC) reflect activation in the right frontal cortex. There was no relationship between natural activation and change scores for bilinguals. Monolinguals exhibited a negative relationship with change scores in optode 14 (right LPFC). This means that as the lag increased between isolation and recognition, the activation level decreased on optode 14 (right LPFC). The second series of analyses correlated consonant voicing features (voiced, voiceless) and vowel voicing features (tense, lax) with neural activation separately for bilinguals and monolinguals for isolation, recognition, and change point scores.

For monolinguals, voiced words in English produced positive relationships between performance and neural activity for isolation on optodes 5 (left MPFC) and 7 (left MPFC). Change scores resulted in a negative correlation for English-Voiced on optode 7 (left MPFC), indicating that as the lag between isolation and recognition increased, neural activation decreased. For bilinguals, faster recognition times correlated with increased activation in optodes 2 (left LPFC) and 5 (left MPFC) in the Spanish-Tense condition. Change scores yielded significant relationships across the frontal cortex for bilinguals in both tense and lax conditions. However, longer lags between isolation and recognition were associated with increased neural activation for tense and decreased neural activation for lax in mostly different areas of the frontal cortex. See Table 5.

## 4. Discussion

The first research question asked whether bilingual speakers identified sounds better in one language vs. the other and also asked whether bilingual speakers are comparable to monolingual native English speakers. Specifically, question one investigated phonotactics between bilingual (early AoA, Spanish-English speakers) and monolingual English speakers across four code-mixed language conditions (i.e., English-English, English-Spanish, Spanish-Spanish, Spanish-English). After Bonferroni corrections, it was found that bilingual speakers (7.70 ms) were faster at identifying English-voiced sounds than monolingual English speakers (13.30 ms) under initial isolation conditions. English-voiced consonants overlap Spanish-voiceless consonants (e.g., /p, t, k/ in voice onset times. English-voiced consonants have a short lag VOT from 0 to 20 msec (Balukas and Koops 2015; Brice and Brice 2008; Piccinini and Arvaniti 2015; Thornburgh and Ryalls 1998); while Spanish-voiceless consonants have a short lag of less than 20 ms (Ryalls et al. 2004). Consequently, the bilingual speakers were able to identify both the English and Spanish consonants and discriminate between the two languages before the monolingual speakers. In summary, bilingual speakers demonstrated abilities in both languages and overall were comparable to monolingual English speakers. Figures 1–4 illustrate the overall abilities of the groups demonstrating the strengths and weaknesses of both bilingual and monolingual speakers.

Regarding the second research question, whether bilingual or monolingual speakers identified sounds better in code-mixed vs. non-mixed sentences, it was also found that bilingual speakers (7.00) were also faster at recognizing Spanish tense words than monolingual speakers (12.70). Spanish does not tend to have lax vowels; hence, bilingual speakers were able to switch off any lax vowels in processing and to just focus on the tense vowels in either Spanish or English. Bilingual speakers were also able to identify voiced words quicker in English perhaps due to Spanish-English overlap in voice onset times. These effects are most likely due to positive transference with bilinguals using both languages over an extended period of time; hence, in some instances, evidence of a bilingual advantage. Overall, bilingual and monolingual speakers were similar.

The third research question investigated neural activation areas between bilinguals and monolingual speakers when identifying words in code-mixed vs. non-mixed sentences. When examining differences for languages (i.e., research question three), there were significant differences between English-English, English-Spanish, Spanish-Spanish, and English-Spanish for isolation and recognition for the monolingual speakers; with trends towards significance for change scores. No such differences were found for the bilingual speakers. Generally speaking, the monolingual speakers identified and recognized code-mixed target words faster in English (i.e., English-English and Spanish-English), and the Spanish speakers identified and recognized the target words in Spanish faster (i.e., English-Spanish and Spanish-Spanish). The change scores indicated the time difference between the isolation scores and recognition scores or the time spent deciding upon the word recognition. For the English-English condition, the change score was almost identical between the two groups. The Spanish-English condition was almost identical between the two groups again. For English-Spanish, the bilingual speakers identified and recognized faster than the monolinguals. In addition, the bilingual speakers demonstrated lower change scores indicating less time deciding upon the word. For the Spanish-Spanish change scores, the monolinguals demonstrated less time deciding upon the word. Although the bilinguals identified and recognized the word quicker, they spent more time making the recognition decision (i.e., a greater change score). This raises the question of a slight bilingual "tax" (i.e., a delay processing cost) for alternating between the two languages. Stasenko et al. (2021) refer to this as a processing cost when alternating languages, i.e., loss of inhibitory control. Inhibitory control is believed to decline with aging. Hence, intensive cognitive-language processing tasks (such as code-mixing during a gating task) may increase episodes of loss of inhibitory control.

The fourth area of investigation examined neural activation areas between bilinguals and monolingual speakers when identifying sounds in code-mixed vs. non-mixed sen-

tences. Research question four examined relationships between the groups on phonotactics and neural activation. For monolinguals under the isolation condition for English-voiced sounds, the medial frontal lobe areas (optodes 5 left MPFC and 7 left MPFC) were significantly activated. It is surmised that this may be due to the overlap between the English /b/ and the Spanish /p/ in VOT, causing some phonotactic delays in processing. This also raises the question of phonotactic interference even if one is monolingual when confronted in bilingual speaking situations. Further research is warranted. The difference in change score was significant with $p \leq 0.05$; however, was not significant with $p \leq 0.01$ ($p = 0.013$). A negative correlation score indicated that the relationship was inverse, i.e., as the lag between the isolation point and recognition point increased (taking longer to make a final decision), neural activation decreased instead of increasing. This appears to have a negative effect on monolinguals' decision-making. Recall that the change score reflects how much time a person took between initially reporting a response and confirming the identification of the word. A short change score may occur early or late in the word presentation. For some participants, they waited until close to the end of the word before reporting an initial response (i.e., recognition score), therefore, quickly confirming their response by the end of the word (i.e., isolation score).

Other participants reported an initial recognition midway through the word and may have confirmed their recognition quickly (i.e., short change score) or waited until the end of the word (i.e., long change score). A short change score could then include both people who were early to report a recognition and those who were late to report a recognition. Regardless of when the participant initiated the recognition, those who quickly verified their recognition yielded larger activations. For those who initiated an early recognition, but took longer to verify their recognition, less neural activation was observed.

The neural efficiency hypothesis indicates that individuals with high cognitive ability require less activation under less demanding tasks and higher activation during more demanding tasks (Causse et al. 2017; Haier et al. 1992; Neubauer and Fink 2009). In this case, less activation is associated with delay in processing or impulsive responding, which does not reflect the neural efficiency hypothesis and instead suggests minimal recruitment of decision-making areas in the prefrontal cortex for long lag times and maximal recruitment of these areas for short lag times. The behavior may be considered impulsive if one makes an early recognition based on partial hearing of the word that must be revised with incoming stimuli rather than deliberately waiting until more of the word has been processed. Therefore, the change score holds value by illuminating more aspects of the relationship with neural activation than the recognition and identification scores alone.

Bilingual speakers demonstrated significance when the phonotactic stimuli included Spanish tense words under recognition scores (optode 4, left LPFC). Recall that Spanish contains few lax vowels resulting in the narrowing of word candidates in Spanish along with Spanish containing fewer vowels than English (Delattre 1965). Neural activation under these conditions involved left hemisphere ventromedial prefrontal lobes. The bilingual speakers showed increased activation in decision-making, which appears to be a language advantage that may have led to faster recognition. However, in Spanish, the bilingual speakers also experienced difficulties with English-lax vowels in making their recognition choice as evidenced by their longer change scores. Recall that lax vowels are not represented in Spanish. Consequently, bilingual speakers displayed longer lags (as indicated by the negative Spearman's rank correlation) with English-lax vowels (see Table 5). Under English-lax phonotactics and change scores, bilingual speakers demonstrated significance for optodes 4 (left LPFC) and 11 (right MPFC). Optodes 4 (left LPFC) corresponds to the left hemisphere for identification and 11 right MPFC) corresponds to the right hemisphere for decision-making. However, the bilingual speakers achieved overall parity with the monolingual speakers (Archila-Suerte et al. 2015). This study also supports the notion that neural processing both overlaps and differs between bilinguals and monolinguals. Further research is suggested for more thorough examinations with larger samples.

*Limitations, Strengths, and Future Directions*

The greatest limitation of this study is the small sample size. It is noted that neuroimaging research tends to have much smaller sample sizes (Szucs and Ioannidis 2020). Szucs and Ioannidis stated that 96% of highly cited fMRI studies employed only one group with a median total sample size of 12. Nevertheless, this study utilized a matching sample to produce a balanced design with an equal number of bilingual and monolingual participants (n = 20). The participants were balanced in terms of age, gender, and ethnicity. To address the sample size, non-parametric analyses were used instead of parametric analyses. Thus, although the sample size was smaller than desired, the conclusions can be generalized and inform future studies.

One strength of the study was that the sample was relatively homogenous. The bilingual participants were all early L2 age of onset (i.e., birth to 8 years of age) with Spanish being the primary language. While the age of onset was well controlled in this study (using only early AoA individuals or early bilinguals), future directions for research should also include comparisons of early age of onset, middle, and later age of onset. This will help evaluate the possibility that, despite spoken proficiency, differences in the onset of L2 could influence how participants perceive the phonetic features of words. While the sample included only bilingual individuals, studying multilingual (i.e., polyglot) individuals could allow for further understanding of the influences of multiple languages on word identification.

An additional source of confounding variables includes the language stimuli employed in this study. Word frequency, word length, and neighborhood density (F. Grosjean, personal correspondence, 8 February 2017) were partially controlled for in the present study. However, efforts were made to control for the context of the carrier sentence by eliminating contextual features that could be used to predict the target word and ensuring all target words were nouns.

A strength of the present study is that all gated words were nouns presented in a neutral context, which allowed us to examine how features other than context affected word recognition. Future research could examine words of similar length (e.g., a sampling of entirely trisyllabic words) and of similar frequency of occurrence (e.g., common words). The present study also examined only bilinguals who were proficient in Spanish and English for the sake of keeping the study design reasonably simple. Future researchers should consider including those who are multilingual, comparing them both to bilingual and monolingual individuals. This could be particularly advantageous in study designs where L1 and L2 are not very closely related (e.g., Cantonese and Spanish as opposed to Spanish and English); it is possible that if multilingual individuals know a wider breadth of languages, word recognition might be improved.

The bilingual participants utilized in this study were speakers of Spanish and English. Future studies could utilize bilingual participants who know different combinations of languages. In particular, this study could be further expanded by studying participants who know languages that utilize writing and phonetic systems that are further differentiated from each other than English and Spanish (e.g., English and Japanese). Likewise, it would be informative to examine differences among participants who know three or more languages and to differentiate balanced bilinguals vs. unbalanced bilinguals (Peltola et al. 2012). Moreover, different measures of neurophysiology could be applied to the experimental paradigm of using the gating task (e.g., EEG).

Finally, it is possible that the experimental design used for this study could be adapted for longitudinal designs. For instance, participants could be initially assessed as young children and then retesting them as emerging adults. Alternatively, a future longitudinal design could examine differences between emerging young adults and older adults (i.e., ages 65+).

## 5. Conclusions

Both the monolingual speakers and the bilingual speakers demonstrated more similarities than differences in the total number of findings as indicated by the number of nonsignificant findings (Archila-Suerte et al. 2015). This would seem to suggest that this parity is due to early AoA bilinguals finding a balance of language capabilities, i.e., native-like proficiency. Figure 3 seems to illustrate this under the English-English (non-mixed) conditions where the bilingual speakers showed less neural activation than monolingual speakers (i.e., requiring less oxygenation and mental effort). This parity is not equal across all phonotactic, language, and neural conditions. The parity includes advantages in some situations and disadvantages in other situations that tend to equal out over all the above-mentioned conditions. This would support Abutalebi et al. (2001), who assert that in some instances L2 processing occurs in the same areas dedicated to first language processing when code-switching. In addition, the findings from this research support the notions that many language centers employ shared networks across the frontal lobes, e.g., both left and right hemispheres.

What is surprising from this study was the comparable performance on Spanish target words for the monolingual speakers. The sample was collected from university students, which limits the generalizability to a broader population. However, these monolingual speakers, although not fluent in Spanish, had been exposed to some levels of Spanish in their environments (e.g., Spanish classes in high school, secondary education; friends who spoke Spanish, etc.). Spanish and English share Indo-European language roots and their similarities are noted (Pagel et al. 2007). In addition, it appears that even minimal second language exposure has a positive bilingual effect on their speech perception (see Tables 4 and 5).

The neural areas of the ventromedial prefrontal lobes as represented by optodes 5 (left MPFC), and 7 (left MPFC) were crucial for monolingual speakers for English-voiced consonants (/b, d, g/). The VOT overlap with Spanish-voiceless consonants may have affected their identification and recognition of these words. In sum, our results support those of Archila-Suerte et al. (2015) and Perani et al. (1998) that localization of speech perception may be similar for both monolingual and bilingual populations, yet their levels of activation differed. Early age of acquisition bilinguals in this study seemed to recruit additional areas when engaging in code-mixing, especially involving the left lateral prefrontal cortex (optode 2, left LPFC). Further research is needed to investigate these issues in what appears to be nascent areas of study. In conclusion, our findings suggest that the aforementioned parity is due to early age of acquisition (AoA) bilinguals finding a balance of language capabilities (i.e., native-like proficiency) and that in some instances the bilingual speakers processed language in the same areas dedicated to first language processing (Abutalebi 2007; Archila-Suerte et al. 2015) indicating overlap of the two languages during early childhood brain development.

**Author Contributions:** Conceptualization, A.E.B. and C.S.; methodology, A.E.B. and C.S.; software, A.E.B. and C.S.; validation, A.E.B., C.S. and M.K.M.; formal analysis, A.E.B., C.S. and M.K.M.; investigation, A.E.B. and C.S.; resources, A.E.B., C.S. and M.K.M.; data curation, A.E.B., C.S. and M.K.M.; writing—original draft preparation, A.E.B.; writing—review and editing, A.E.B., C.S. and M.K.M.; visualization, A.E.B.; supervision, A.E.B.; project administration, A.E.B. All authors have read and agreed to the published version of the manuscript.

**Funding:** This research received no external funding.

**Institutional Review Board Statement:** The authors followed all Institutional Review Board (IRB) policies and procedures at the university where the research was conducted. The authors also report no declaration of interest. The authors have no financial interest, direct or indirect, in the subject matter or materials discussed in the manuscript. This study was funded by a University of South Florida Creative Scholarship Grant.

**Informed Consent Statement:** Informed consent was obtained from all subjects involved in the study.

**Conflicts of Interest:** The authors declare no conflict of interest.

## Appendix A

**Table A1.** Stimulus Carrier Phrase and Target Words (Translations in parentheses).

| Voiced Initial CC | Unvoiced Initial CC |
|---|---|
| 1. He is bruto (He is stupid). | 1. What is your problema? (What is your problem?) |
| 2. He is bright. | 2. What is your progress? |
| 3. I see a bruja (I see a witch). | 3. Give me the plata (Give me the money/silver). |
| 4. I see a broom. | 4. Give me the plate. |
| 5. Dame el brush (Give me the brush). | 5. Yo quiero cheese (Y want cheese). |
| 6. Dame el brillo (Give me the brightness). | 6. Yo quiero chicle (I want gum). |
| 7. Y vi un dragon (I see a dragon). | 7. Is that my problema? (Is that my problem?) |
| 8. Yo vi un drama (I saw the drama). | 8. Give me the chicle (Give me the gum). |
| 9. I saw the drama. | 9. Quiero un plate (I want a plate). |
| 10. Eso es green (That is green). | 10. Eso es cheese (That is cheese). |
| 11. Eso es grande (That is big). | 11. Necesito un plate (I need a plate). |
| 12. That is gratis (That is free). | 12. I have a chicle (I have gum). |
| 13. That is great. | 13. Is that my plata? (Is that my money/silver?) |
| 14. He is grande (He is big). | 14. Dame el plate. (Give me the plate). |
| 15. Dame el brazo (Give me the arm). | 15. Dame el cheese (Give me the cheese). |
| 16. I see a brillo (I see brightness). | 16. I have a problema (I have a problem). |
| 17. I see the bruja (I see the witch). | 17. Yo vi un plate (I saw a plate). |
| 18. Give me the brillo (Give me the brightness). | |
| **Tense Initial CV** | **Lax Initial CV** |
| 1. Give me the carta (Give me the letter). | 1. Give me the card. |
| 2. Is that my café? (Is that my coffee)? | 2. Is that my car? |
| 3. I made a lísta (I made a list). | 3. Yo vi el cartoon (I saw the cartoon). |
| 4. I made a leash. | 4. Vi el poster (I saw the poster). |
| 5. Dame el leash (Give me the leash). | 5. Necesito un cup (I need a cup). |
| 6. I have a libro (I have a book). | 6. I have a leaf. |
| 7. I saw the muñeca (I saw the doll). | 7. Yo vi el car (I saw the car). |
| 8. I saw the moon. | 8. Necesito un car (I need a car). |
| 9. Yo vi el carro (I saw the car). | 9. Quiero un cup (I want a cup). |
| 10. Quiero un cuchillo (I want a knife). | 10. Yo vi un poster (I saw a poster). |
| 11. Quiero un cookie (I want a cookie). | |
| 12. Queiro un lollipop (I want a lollipop). | |
| 13. Quiero un lapiz (I want a pencil). | |
| 14. Vi el postre (I saw the poster) | |
| 15. Necesito un corte (I need a cut). | |

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
