# Peer review of "Neural Activation in Bilinguals and Monolinguals Using a Word Identification Task"

_languages, doi:10.3390/languages8030216_

Round 1

Reviewer 1 Report

This is study investigates the neural processing of language, with a focus on phonotactics in early Spanish-English bilinguals while they listened to code-mixed stimuli in Spanish and English as compared to monolinguals, using a gating paradigm. The study packs in a number of important variables in order to get a closer look at bilingual processing in a codeswitching (specifically code-mixing) context, an understudied but crucial area in bilingual processing. While the content of the study is important and informative, following are a number of suggestions to help the content of the paper better. 

  1. The article could be stronger if it BEGAN with a statement of problem. This would be particularly helpful because the study spans a number of variables and the introduction is fairly broad, even after 1-2 pages, it is hard for the reader to understand what the study is going to focus on. A succinct initial paragraph with the statement of problem, stating existing research and the gap in the literature followed by a clear rationale for the study ending in the purpose of the study would give the reader some idea, going into the rest of the introduction, how to process the literature review that is presented. 

  1. The introduction (literature of review) is very broad and could benefit from some tightening and better organization so that the flow from section to section is easier for the reader to follow and understand how it leads up to the study. The end of the introduction would read better as a summary bringing all the variables together in a succinct paragraph before the research questions. Here are other important comments on the introduction: 

  1. A discussion in the introduction addressing Age of Arrival and how that is different from Age of Acquisition would be important since this is a long standing variable in the Second Language Acquisition literature 

  1. P6 L276-286 are a repetition of the L216-226 on P5. 

  1. The 4 “research questions” as presented read as hypotheses instead of research questions. It would help the readers if the authors presented the research questions first followed by the hypotheses. 

  1. In the section labeled research questions, number 1 refers to phonotactic differences, phonotactics of which language is being alluded to here?  

  1. In the section labeled research questions, number 2 mentions “....four code-mixed language conditions...” and mentions English-English and Spanish-Spanish as the part of the code-mixed language conditions, but only the English-Spanish and Spanish-English are the only code-mixed ones, to my understanding?  

  1. The one large question about the methods is that it is not clear why the authors chose to test monolingual individuals on code-mixed stimuli in Spanish and English. 

  1. Here are other important comments on the methods: 

  1. The procedures can be clarified some more and more details and clearer writing will make it easier to follow. 

  1. A figure dedicated to the gating and how that interacted with words in the sentence is crucial to the understanding and data analysis in the study 

  1. How did the experimenters communicate and instruct the participants, in English, Spanish or in both languages?  

  1. What language were the carrier phrases in the code-mixed stimuli?  

  1. Were the stimuli presented only in auditorily? What did the participants do – i.e., was there a fixation on a screen for the participants to look at?  

  1. Where was the experimenter located during the experiment to be able to hear the verbal responses from the participants?  

  1. Were the verbal responses of the participants recorded? 

  1. How long was the fNIRS preparation period, were the electrode recordings happening at the same time as the behavioral responses were being provided by the participants?  

  1. Since the recognition points varied from stimulus to stimulus, was it the case that the participants never heard the full stimuli for all 60 stimuli- i.e. for the one that were fully recognized before the complete stimuli were presented?  

  1. The authors tried providing many details about the analyses but because of the complexity and the number of variables examined, the analyses are not easy to access as they have been presented. More clarity regarding the data analysis in conjunction with the figure explaining gating would be very helpful. 

  1. The results were presented in great detail but certain things remain unclear. If the authors would address the following, it would be very helpful to the reader: 

  1. What do the columns under Monolingual and Bilingual in Tables 1-4 represent 

  1. Results section 3.1.1. is labeled as “within-subject” differences but the first sentence under this heading reads as (L480) “…..between monilinguals and bilinguals” 

  1. The significance of data for monolinguals in the code-mixed Spanish English conditions with Spanish sentences or Spanish targets is not clear 

  1. Figures 2-5 showing the fNIRS data could be made clearer if the optodes with significant differences are marked. It isn’t clear why figure 2 shows a large difference (in fact the largest in all the figures) but isn’t significant, are there outliers? 

  1. The first paragraph in section 3.3. in Results ends in the sentence “No other significant differences were found…” indicating other significant differences were mentioned in the paragraph before, but no statistical differences are mentioned. It may be that the authors are referring to the differences in the Tables, but clarifying this would make it easier to read. 

  1. The Discussion can benefit from stating the actual research questions while discussing the results 

  1. Because the relationship between optodes and brain regions are not frequent in the literature, brain regions when mentioning the optodes is useful to the reader. A schematic of the optodes on the head to understand the positions would also be useful. 

  1. Other minor requests are listed below: 

  1. P1 L3-31: “Thus perceiving….” could benefit from rephrasing as this sentence is difficult to understand. 

  1. P L39-40 mentions “These developmental learning differences between L1 and L2…but L1 and its characteristics are not mentioned before this sentence, so the difference is not apparent. 

  1. P2 L53-56 – Can be clarified more, and the information in the parenthesis need not be presented in parenthesis as it is important information for the sentence and the paragraph. 

  1. P2 L59-61, is the similarity to monolinguals mentioned here true in both languages?  

  1. P2 L72: “Thus it is more likely that factors..” - it is not clear why it is MORE LIKELY?  

  1. P3 L120: “Lax vowels. Spanish does not....” - Perhaps this sentence is placed in error? 

  1. P4 L159-183: This section starting with “Bric and Brice (2008)…" addressing speech perception in bilinguals in general – not pertaining to code-switching- reads abruptly. 

  1. P4 L249-253: The first sentence of this paragraph reads abruptly, the words “...did differ...” in the sentences reads as if something specific in a study is being referred to – not the general notion of speech perception differences between bilinguals and monolinguals.  

  1. P7-8 L354-355: Would it make sense to mention that the proficiency mentioned here refers to the proficiency measures mentioned in the next section?  

Author Response

Reviewer One

This is study investigates the neural processing of language, with a focus on phonotactics in early Spanish-English bilinguals while they listened to code-mixed stimuli in Spanish and English as compared to monolinguals, using a gating paradigm. The study packs in a number of important variables in order to get a closer look at bilingual processing in a codeswitching (specifically code-mixing) context, an understudied but crucial area in bilingual processing. While the content of the study is important and informative, following are a number of suggestions to help the content of the paper better.

Comment:

  1. The article could be stronger if it BEGAN with a statement of problem. This would be particularly helpful because the study spans a number of variables and the introduction is fairly broad, even after 1-2 pages, it is hard for the reader to understand what the study is going to focus on.

A succinct initial paragraph with the statement of problem, stating existing research and the gap in the literature followed by a clear rationale for the study ending in the purpose of the study would give the reader some idea, going into the rest of the introduction, how to process the literature review that is presented.

Correction: The purpose statement has been moved to the beginning of the manuscript to offer clarity.

Comment

  1. The introduction (literature of review) is very broad and could benefit from some tightening and better organization so that the flow from section to section is easier for the reader to follow and understand how it leads up to the study. The end of the introduction would read better as a summary bringing all the variables together in a succinct paragraph before the research questions. Here are other important comments on the introduction:

Correction: The purpose statement has been moved beginning of the manuscript. Additional text has been added. The subheadings have been adjusted for a better topic flow. All subheadings are aligned with each topic area with improved transitions between topics and subheadings.

Comment:

2.1 A discussion in the introduction addressing Age of Arrival and how that is different from Age of Acquisition would be important since this is a long standing variable in the Second Language Acquisition literature.

Correction: Several sentences have been included to differentiate age of arrival vs. age of acquisition. Age or acquisition is now used consistently in the manuscript.

Comment:

2.2 P6 L276-286 are a repetition of the L216-226 on P5.

Correction:  This has been corrected.

Comment:

  1. The 4 “research questions” as presented read as hypotheses instead of research questions. It would help the readers if the authors presented the research questions first followed by the hypotheses.

Correction: Research questions have been posed prior to the hypotheses.

Comment:

3.1  In the section labeled research questions, number 1 refers to phonotactic differences, phonotactics of which language is being alluded to here?

Correction: Both languages. This has been clarified.

Comment: 

3.2  In the section labeled research questions, number 2 mentions “ ... four code- mixed

language conditions . ” and mentions English-English and Spanish-Spanish as the part of

the code-mixed language conditions, but only the English-Spanish and Spanish-English are

the only code-mixed ones, to my understanding?

Correction: Correct. This has been clarified.

Comment:

  1. The one large question about the methods is that it is not clear why the authors chose

to test monolingual individuals on code-mixed stimuli in Spanish and English.

Correction: See research question one. In addition, the issue of native-like abilities among

bilingual speakers cannot be made unless compared to monolingual speakers of that

language. This has been clarified.

Comment:

5 .     Here are other important comments on the methods:

5.1  The procedures can be clarified some more and more details and clearer writing will make it easier to follow.

Correction: The Procedures section has been written for better detail.

Comment:

5.2  A figure dedicated to the gating and how that interacted with words in the sentence is crucial to the understanding and data analysis in the study.

Correction: This is clarified under Materials and Methods, Language stimuli. All gated words occurred at the end of a carrier phrase. Carrier phrases were general only to indicate top-down information that the words was a “noun”. The list of stimuli words are provided in Appendix A.

Comment:

5.3  How did the experimenters communicate and instruct the participants, in English, Spanish or in both languages?

Correction: This has been clarified further. Both languages for bilingual participants and English for the monolingual participants. This is indicated Materials and Methods under the subheading of Language proficiency rating.

Comment:

  1. What language were the carrier phrases in the code-mixed stimuli?

Correction: Clarified under Materials and Methods, Language stimuli.

Comment:

  1. Were the stimuli presented only in auditorily? What did the participants do – i.e., was there a fixation on a screen for the participants to look at?

Correction: This information has been added under Procedure.

Comment:

  1. Where was the experimenter located during the experiment to be able to hear the verbal responses from the participants?

Correction: Seated in the same room. This has been clarified under Materials and Methods, Procedure.

Comment:

  1. Were the verbal responses of the participants recorded?

Correction: No. This has been clarified under Materials and Methods, Procedure.

Comment:

  1. How long was the fNIRS preparation period, were the electrode recordings happening at the same time as the behavioral responses were being provided by

the participants?

Correction: This has been clarified under Materials and Methods, Procedure.

Comment:

  1. Since the recognition points varied from stimulus to stimulus, was it the case that the participants never heard the full stimuli for all 60 stimuli- i.e. for the one that were fully recognized before the complete stimuli were presented?

Correction: Not hearing the full stimuli was the intended outcome. It was rare that participants reached the end of the stimuli without a recognition point. This has been clarified under Materials and Methods, Procedure.

Comment:

  1. The authors tried providing many details about the analyses but because of the complexity and the number of variables examined, the analyses are not easy to access as they have been presented. More clarity regarding the data analysis in conjunction with the figure explaining gating would be very helpful.

Correction: The data analysis has been clarified.

The results were presented in great detail but certain things remain unclear. If the authors would address the following, it would be very helpful to the reader:

Comment:

  1. What do the columns under Monolingual and Bilingual in Tables 1-4 represent?

Correction: Answer- the time it took, in milliseconds, to respond in isolating or recognizing the word. This has been clarified with Tables 1-3.

Table 4 indicates fNIRS optodes, correlations, and probabilities associated with the different sentence conditions of code-mixed and non-code-mixed sentences.

Comment:

  1. Results section 3.1.1. is labeled as “within-subject” differences but the first sentence under this heading reads as (L480) “.....between monilinguals and

bilinguals”

Correction: The subheading was incorrect and has been changed.

Comment:

  1. The significance of data for monolinguals in the code-mixed Spanish English conditions with Spanish sentences or Spanish targets is not clear

Correction: Tables 1-3 have further information regarding data values. The data reported in Tables 1-3 report response times in milliseconds; while, Tables 4-5 report indicate optode numbers with corresponding brain areas (e.g., optode 11 right MPFC).

Comment:

  1. Figures 2-5 showing the fNIRS data could be made clearer if the optodes with significant differences are marked. It isn’t clear why figure 2 shows a large difference (in fact the largest in all the figures) but isn’t significant, are there outliers?

Correction: The new Figure 3 (old Figure 2) demonstrates a large difference; yet, these results were non-significant. The authors believe that non-significance resulted from a small sample size.  This is noted in the text, Results, Neural activation with fNIRS.

Comment:

  1. The first paragraph in section 3.3. in Results ends in the sentence “No other significant differences were found...” indicating other significant differences were mentioned in the paragraph before, but no statistical differences are mentioned. It may be that the authors are referring to the differences in the Tables, but clarifying this would make it easier to read.

Correction: The reference to the new Table 1 in text has been corrected. A new Table 1 is provided. All tables and numeration have been corrected.

Comment:

  1. The Discussion can benefit from stating the actual research questions while discussing the results.

Correction: The research questions have been integrated into the Discussion.

Comment:

  1. Because the relationship between optodes and brain regions are not frequent in the literature, brain regions when mentioning the optodes is useful to the reader. A schematic of the optodes on the head to understand the positions would also be useful.

Correction: This is clarified in Table 5 and text. This is provided in Results, Neural Activation with fNIRS subheading. The schematic for the optodes to the head is provided in Figure 1.

Comment:

  1. Other minor requests are listed below:

P1 L3-31: “Thus perceiving....” could benefit from rephrasing as this sentence is difficult to understand.

Correction:

The sentence has been reworded.

Comment:

  1. L39-40 mentions “These developmental learning differences between L1 and L2...but L1 and its characteristics are not mentioned before this sentence, so the difference is not apparent.

Correction: The sentence has been clarified.

Comment:

  1. P2 L53-56 – Can be clarified more, and the information in the parenthesis need not be presented in parenthesis as it is important information for the sentence and the paragraph.

Correction: I believe that the reference to the information in parentheses is part of a quote.

Comment:

  1. P2 L59-61, is the similarity to monolinguals mentioned here true in both languages?

Correction: The mention to the native language has been included. See reference to Birdsong, (2018).

Comment:

  1. P2 L72: “Thus it is more likely that factors..” - it is not clear why it is MORE LIKELY?

Correction: This has been clarified with an additional source.

Comment:

  1. P3 L120: “Lax vowels. Spanish does not. ..” - Perhaps this sentence is placed in error?

Correction: This has been corrected.

Comment:

  1. P4 L159-183: This section starting with “Bric and Brice (2008)..." addressing speech perception in bilinguals in general – not pertaining to code-switching- reads abruptly.

Correction: This sentence has been cleaned.

Comment:

  1. P4 L249-253: The first sentence of this paragraph reads abruptly, the words “...did differ..” in the sentences reads as if something specific in a study is being referred to – not the general notion of speech perception differences between bilinguals and monolinguals.

Correction: The first sentence has been changed.

Comment:

  1. P7-8 L354-355: Would it make sense to mention that the proficiency mentioned here refers to the proficiency measures mentioned in the next section?

Correction: Yes, this has been mentioned.

Reviewer 2 Report

The results look great and Figure 2 is very impressive.

Here are some suggestions to help improve the manuscript.

1) Show a diagram/photograph of the fNIRS equipment.

2) Show example fNIRS signal are different stages of processing.

3)  Add a spearman correlation between fNIRS  brain language score and behavioral language score for the different subject groups.  Show a plot of this correlation.

Author Response

Reviewer Two

The results look great and Figure 2 is very impressive.

Here are some suggestions to help improve the manuscript.

Comment:

1) Show a diagram/photograph of the fNIRS equipment.

Correction: Please refer to the new Figure 1.

Comment:

2) Show example fNIRS signal are different stages of processing.

Correction: Refer to the new Figure 2.

Comment:

3) Add a spearman correlation between fNIRS brain language score and behavioral language score for the different subject groups. Show a plot of this correlation.

Correction: Tables 5 provides the Spearman rank correlations for optodes (fNIRS) and language conditions;  Table 6 provides Spearman rho rank correlations for optodes (fNIRS) and phonotatic conditions. The Figures provide a plot for these interactions as the Language groups (monolingual and bilingual) are correlated versus fNIRS optodes. Each figure illustrates each of the sentence-word conditions: 1. English-English; 2. English-Spanish; 3. Spanish-Spanish; and, 4. Spanish-English.